# The co-occurrence of multimorbidity and polypharmacy among middle-aged and older adults in Canada: A cross-sectional study using the Canadian Longitudinal Study on Aging (CLSA) and the Canadian Primary Care Sentinel Surveillance Network (CPCSSN)

Kathryn Nicholson[1,2☯]*, Jennifer Salerno[2,3☯], Sayem Borhan[3‡], Benoit Cossette[4‡], Dale Guenter[2‡], Meredith Vanstone[2‡], John Queenan[5‡], Michelle Greiver[6‡], Michelle Howard[2‡], Amanda L. Terry[1,7,8‡], Tyler Williamson[9‡], Lauren E. Griffith[3‡], Martin Fortin[10‡], Saverio Stranges[1,7,11,12‡], Dee Mangin[2,13☯]

1 Department of Epidemiology & Biostatistics, Western University, London, Ontario, Canada, 2 Department of Family Medicine, McMaster University, Hamilton, Ontario, Canada, 3 Department of Health Research Methods, Evidence and Impact, McMaster University, Hamilton, Ontario, Canada, 4 Faculté de médecine et des sciences de la santé, Université de Sherbrooke, Sherbrooke, Québec, Canada, 5 Department of Family Medicine, Queen's University, Kingston, Ontario, Canada, 6 Department of Family and Community Medicine, University of Toronto, Toronto, Ontario, Canada, 7 Department of Family Medicine, Western University, London, Ontario, Canada, 8 Interfaculty Program in Public Health, Western University, London, Ontario, Canada, 9 Department of Community Health Sciences, University of Calgary, Calgary, Alberta, Canada, 10 Department of Family Medicine and Emergency Medicine, Université de Sherbrooke, Sherbrooke, Québec, Canada, 11 Department of Medicine, Western University, London, Ontario, Canada, 12 Department of Clinical Medicine and Surgery, University of Naples Federico II University, Naples, Italy, 13 Department of Primary Care and Clinical Simulation, University of Otago, Christchurch, New Zealand

☯ These authors contributed equally to this work.
‡ SB, BC, DG, MV, JQ, MG, MH, ALT, TW, LEG, MF and SS also contributed equally to this work.
* kathryn.nicholson@schulich.uwo.ca

## Abstract

### Background

There is an increasing prevalence of multiple conditions (multimorbidity) and multiple medications (polypharmacy) across many populations. Previous literature has focused on the prevalence and impact of these health states separately, but there is a need to better understand their co-occurrence.

### Methods and findings

This study reported on multimorbidity and polypharmacy among middle-aged and older adults in two national datasets: the Canadian Longitudinal Study on Aging (CLSA) and the Canadian Primary Care Sentinel Surveillance Network (CPCSSN). Using consistent methodology, we conducted a cross-sectional analysis of CLSA participants and CPCSSN patients aged 45 to 85 years as of 2015. When multimorbidity was defined as two or more conditions, the prevalence was 66.7% and 52.0% in the CLSA and CPCSSN cohorts,

**Data Availability Statement:** Data are available from the Canadian Longitudinal Study on Aging (www.clsa-elcv.ca) for researchers who meet the criteria for access to de-identified CLSA data. Data are available from the Canadian Primary Care Sentinel Surveillance Network (https://cpcssn.ca/) for researchers who meet the criteria for access to de-identified CPCSSN data.

**Funding:** This study was supported by the Canadian Institutes of Health Research Priority Announcement in Aging (175365). DM received the funding from CIHR for this project. This funding source had no role in the conduct of this study, the writing of the manuscript or the decision to submit for publication. Funding for the CLSA is provided by the Government of Canada through the Canadian Institutes of Health Research (CIHR) under grant reference: LSA 94473 and the Canada Foundation for Innovation, as well as the following provinces, Newfoundland, Nova Scotia, Quebec, Ontario, Manitoba, Alberta, and British Columbia. CPCSSN is funded through a variety of sources such as the Canadian Institutes for Health Research, Departments of Family Medicine and the Canadian Institute for Military and Veteran Health Research. There was no additional external funding received for this study.

**Competing interests:** The authors have no conflict of interest to disclose. The opinions expressed are the authors' own and do not reflect the views of the Canadian Longitudinal Study on Aging (CLSA) or the Canadian Primary Care Sentinel Surveillance Network (CPCSSN).

respectively. The prevalence of polypharmacy was 14.9% in the CLSA cohort and 22.6% in the CPCSSN cohort when defined as five or more medications. Using the same cut-points, the co-occurrence of multimorbidity and polypharmacy was similar between the two cohorts (CLSA: 14.3%; CPCSSN: 13.5%). Approximately 20% of older adults (65 to 85 years) were living with both multimorbidity and polypharmacy (CLSA: 21.4%; CPCSSN: 18.3%), as compared to almost 10% of middle-aged adults (45 to 64 years) living with this co-occurrence (CLSA: 9.2%; CPCSSN: 9.9%). Across both cohorts and age groups, females had consistently higher estimates of multimorbidity, polypharmacy and the co-occurrence of multimorbidity and polypharmacy.

## Conclusions

This study found that multimorbidity and polypharmacy are not interchangeable in understanding population health needs. Approximately one in five older adults in the CLSA and CPCSSN cohorts were living with both multimorbidity and polypharmacy, double the proportion in the younger cohorts. This has implications for future research, as well as health policy and clinical practice, that aim to reduce the occurrence and impact of multimorbidity and unnecessary polypharmacy to enhance the well-being of aging populations.

## Introduction

The advances of public health and clinical medicine have allowed individuals to live longer, but similar increases in quality of life have not followed [1]. While an increased lifespan represents an important achievement, there is now a need to ensure that health span (quality of life) extends along with this lifespan (quantity of life). The accumulation of multiple conditions (multimorbidity) within an individual, as well as the multiple medications (polypharmacy) used to manage or treat these conditions, have become significant challenges for public health, health care systems, health care providers and individuals living with these health states [2–7]. Multimorbidity and polypharmacy are often strongly associated [8–10]. A powerful driver of this is the current clinical guidelines framework that is typically used in providing care [2, 4, 7]. These have been largely developed for single conditions in isolation, yet do not sufficiently account for common scenarios in which management of multiple conditions may be required at one time [6, 11–16]. Indeed, addressing the use of multiple prescribed medications and minimizing the potential occurrence of harm and treatment burden have been acknowledged as key tasks for improvement in the clinical management of multimorbidity [3, 4, 6, 17–21].

To date, there have been a number of reviews conducted to describe the measurement, prevalence and factors associated with the occurrence of multimorbidity and polypharmacy separately in various settings [4, 22–38]. Living with multimorbidity and polypharmacy can create substantial challenges that extend beyond the conditions and symptoms themselves [6, 34, 39] including increased risk of medication interactions [3, 4, 15, 34, 40, 41], barriers to self-care and self-management [42–45], reduced quality of life [32, 35, 43, 46, 47], higher healthcare costs [48–52] and difficulty coordinating health care services [7, 43, 53–55]. Polypharmacy also carries the potential for inappropriate or problematic prescribing, particularly as the number of concomitant medications increases, if these are not managed and reviewed regularly by health care providers, patients and caregivers [31, 56].

A recent systematic review was conducted to identify original research that reported on the prevalence of both multimorbidity and polypharmacy among adults or older adults in community or primary care settings [57]. The 87 included studies reported a prevalence of multimorbidity (defined as two or more conditions) that ranged from 4.8% to 93.1%, while the reported prevalence of polypharmacy (defined as five or more medications) ranged from 2.6% to 86.6% [57]. Of course, these substantial ranges in prevalence are partly due to the methodological heterogeneity and the lack of a consistent operational definition for these two concepts [4, 5, 30, 36–38], but this review also identified the lack of studies that reported on the prevalence of the co-occurrence of multimorbidity and polypharmacy within the same individuals (that is, instead of the calculation and reporting on multimorbidity and polypharmacy separately). Based on previous literature that has highlighted the higher risk of negative health outcomes (such as drug-drug interactions, drug-disease interactions and hospital admissions) for individuals living with both multimorbidity and polypharmacy, there is a need to identify and describe the prevalence of these concurrent health states in various populations.

However, there is no single set of data that represents the "gold standard" approach to reporting on multimorbidity and polypharmacy. For example, clinical and administrative data may capture medications that have been prescribed or dispensed, but health survey data may capture medications that are actually being taken regularly by individuals. Previous research has shown that self-reported data has reasonable levels of validity and accuracy when compared with clinical or administrative datasets, but this can vary by specific conditions [58–60]. Self-reported medication data show reasonable accuracy when compared to prescribing records, particularly for medications that are being used to treat chronic conditions [61–63]. For both conditions and medications, self-reported data can be influenced by factors such as age, gender, health literacy, recall bias and social desirability bias. The accuracy and validity of clinical data for conditions and medications requires consistent and complete documentation by primary care providers, as well as data processing and standardization, particularly because these data are documented for clinical and not research purposes [64–67]. Linked datasets that combine multiple sources of information can address the limitations of individual sources, but when this is not available, complementary data sources can be used to provide insight into this complex issue.

This study aimed to report on the co-occurrence of multimorbidity and polypharmacy among middle-aged and older adults using consistent methodology in two national datasets in Canada. Our objectives were to: 1) determine the prevalence of multimorbidity and polypharmacy (individually and concurrently) in a national community-based cohort study composed of self-reported data; and 2) determine the prevalence of multimorbidity and polypharmacy (individually and concurrently) in a national electronic medical record dataset composed of primary care data.

## Methods

### Data sources

The Canadian Longitudinal Study on Aging (CLSA) is a national prospective longitudinal cohort study collecting data from more than 50,000 community-dwelling participants who were aged 45 to 85 years at recruitment [68–70]. These data are collected either through telephone interviews (Tracking cohort) or directly in-person at the participants' home or at data collection sites (Comprehensive cohort) that are located across the country with follow-up data collection occurring every three years [68–70]. The Comprehensive cohort was composed of participants who lived within 25 to 50 km of one of the eleven data collection sites located in seven provinces and baseline data collection was completed as of 2015 [68–70]. Due to the in-

person and more in-depth data collection, the Comprehensive cohort may have an underrepresentation of individuals with lowers levels of literacy in French or English, memory impairment or mobility issues [69, 70]. However, this cohort had information recorded for both conditions and medications. To capture multimorbidity, the participants were asked "Has a doctor ever told you that you have. . ." for each of the conditions in the multimorbidity operational definition. To capture polypharmacy, the participants were asked to present all regularly scheduled or taken medications, which were then mapped to the World Health Organization (WHO) Anatomical Therapeutic Chemical (ATC) classification [68, 71]. The CLSA data were initially accessed on 02/02/2022 and no individual participants could be identified within the dataset used for research purposes.

The Canadian Primary Care Sentinel Surveillance Network (CPCSSN) is a national electronic medical record (EMR) database collecting longitudinal data from more than 1.5 million community-dwelling patients who are receiving care from a participating primary care site [72–75]. These data are collected from all patient encounters with a primary care provider who has consented to contributing de-identified data to the centralized dataset, which are then cleaned and coded by algorithms to facilitate use [73, 76]. These data are compiled into this centralized dataset from ten regional networks with primary care sites from inner-city, urban, suburban, small town and rural settings [73]. This database has been shown to have a reasonable level of representativeness and is continuously working to expand its coverage [73, 77]. A two-year contact group was used to identify the sample of patients who had at least one primary care encounter over a two-year timeframe (January 1, 2014 to December 31, 2015) and who were more likely to have up-to-date documentation [73]. The conditions and medications that were documented between 2010 and 2015 were used to identify multimorbidity and polypharmacy. More specifically, the date on which a condition was diagnosed or a medication was prescribed was between 2010 and 2015 for reasonable levels of data quality and completeness [73]. The CPCSSN data were initially accessed on 18/07/2022 and no individual patients could be identified within the dataset used for research purposes.

## Study samples

For the purpose of comparison between the two datasets, the sample for this study focused on CLSA participants and CPCSSN patients aged 45 to 85 years as of 2015. In the CLSA cohort, this included individuals who were recruited at baseline and had complete data on birth year and sex. In the CPCSSN cohort, this included individuals who had at least one primary care encounter over a two-year period (2014 to 2015) who had complete data on birth year and sex. The denominator for the CLSA dataset was all participants aged 45 to 85 years as of 2015 in the Comprehensive baseline cohort (N = 30097) and the denominator for the CPCSSN dataset was all patients aged 45 to 85 years as of 2015 who had an encounter with their primary care provider between 2014 and 2015 (N = 597631). Each of these are unique participants and patients, and there is a possibility that the same individuals are included in the CLSA and CPCSSN datasets, but no current linkage can confirm this.

## Definition of multimorbidity

Multimorbidity was identified using a list of 18 conditions and two cut-points (two or more and three or more conditions). Although the original list consists of 20 conditions, 18 of the 20 conditions were used in the current study to align with the conditions captured in the CLSA dataset and the corresponding International Classification of Disease (ICD-9) codes that were applied in the CPCSSN dataset [78]. The list of conditions and the corresponding ICD-9 codes is presented in S1 Table.

### Definition of polypharmacy

Polypharmacy was identified using ATC Level 4 codes and two cut-points (five or more and ten or more medications). There was a total of 909 codes within Level 4 and if multiple medications were being taken within the same ATC Level 4, this was only counted once for a participant or patient (for example: multiple corticosteroids were only counted once as C05AA). The list of included ATC codes is presented in S2 Table.

### Statistical analysis

The mean number of conditions, mean number of medications and the distribution of participants or patients using the two cut-points of multimorbidity (two or more and three or more conditions) and polypharmacy (five or more and ten or more medications) were reported for the overall CLSA and CPCSSN cohorts, as well as stratified by age group (45 to 64 years or 65 to 85 years) and sex (female or male). More specifically, participants and patients were categorized with neither multimorbidity nor polypharmacy, multimorbidity only, polypharmacy only or both multimorbidity and polypharmacy. The prevalence of the individual conditions and medications that were included in the operational definitions of multimorbidity and polypharmacy were also reported for the overall CLSA and CPCSSN cohorts. As this study was focused on the comparison between two national cohorts using descriptive analyses, no weighting or multivariable analyses were conducted. Only participants and patients with a documented birth year and sex were included in the final samples for each cohort. Data management and data analyses were conducted using Stata SE 17.0 [79].

### Ethics approval

The research ethics approval was obtained from the Hamilton Integrated Research Ethics Board (HiREB) Project Number 14045.

## Results

### Overall samples

The characteristics and prevalence estimates for the overall CLSA and CPCSSN cohorts are presented in Table 1. In both cohorts, the majority of the sample was between 45 and 64 years of age (CLSA: 58.0%; CPCSSN: 57.2%) and female (CLSA: 50.9%; CPCSSN: 55.0%). The prevalence of MM2+ was 66.7% among CLSA participants and 52.0% among CPCSSN patients, while the prevalence of MM3+ was 46.1% in the CLSA cohort and 30.2% in the CPCSSN cohort. Among CLSA participants, the prevalence of PP5+ and PP10+ was 14.9% and 1.4%, respectively. In comparison, the prevalence of PP5+ was 22.6% and the prevalence of PP10 + was 9.8% among CPCSSN patients. When the two cohorts were categorized based on the co-occurrence of multimorbidity and polypharmacy, 14.3% of CLSA participants and 13.5% of CPCSSN patients were living with MM2+ and PP5+ (that is, both multimorbidity and polypharmacy). The most common conditions between the two cohorts were hypertension (CLSA: 36.9%; CPCSSN: 34.7%), obesity (CLSA: 29.7%; CPCSSN: 42.9%) and musculoskeletal problem (CLSA: 27.9%; CPCSSN: 38.7%). The most common medications between the two cohorts were HMG CoA reductase inhibitors (CLSA: 21.0%; CPCSSN: 12.5%), thyroid hormones (CLSA: 12.3%; CPCSSN: 6.0%) and proton pump inhibitors (CLSA: 11.7%; CPCSSN: 13.4%). The frequencies of all conditions and medications that were included in the operational definitions of multimorbidity and polypharmacy are presented in S3 Table.

**Table 1. Characteristics and prevalence of multimorbidity and polypharmacy in CLSA and CPCSSN.**

| | CLSA (N = 30097) | | CPCSSN (N = 597631) | | p-value | |
|---|---|---|---|---|---|---|
| **Age Group** | n | % | n | % | | |
| **45 to 64 Years** | 17451 | 58.0 | 341732 | 57.2 | 0.04 | |
| **65 to 85 Years** | 12646 | 42.0 | 255899 | 42.8 | 0.08 | |
| **Sex** | | | | | | |
| **Female** | 15320 | 50.9 | 328741 | 55.0 | <0.01 | |
| **Male** | 14777 | 49.1 | 268890 | 45.0 | <0.01 | |
| **Mean Conditions (SD), Range** | 2.6 (2.0) | 0–15 | 1.9 (1.6) | 0–14 | | |
| **Mean Medications (SD), Range** | 2.2 (2.4) | 0–21 | 3.0 (5.1) | 0–88 | | |
| **Multimorbidity (MM)** | | | | | | |
| **Two or More Conditions (MM2+)** | 20060 | 66.7 | 310744 | 52.0 | <0.01 | |
| **Three or More Conditions (MM3+)** | 13861 | 46.1 | 180536 | 30.2 | <0.01 | |
| **Polypharmacy (PP)** | | | | | | |
| **Five or More Medications (PP5+)** | 4492 | 14.9 | 135261 | 22.6 | <0.01 | |
| **Ten or More Medications (PP10+)** | 405 | 1.4 | 58821 | 9.8 | <0.01 | |
| **Co-Occurrence of MM2+ and PP5+** | | | | | | |
| **Neither MM2+ Nor PP5+** | 9851 | 32.7 | 232362 | 38.9 | <0.01 | |
| **MM2+ Only** | 15754 | 52.3 | 230008 | 38.5 | <0.01 | |
| **PP5+ Only** | 186 | 0.6 | 54525 | 9.1 | <0.01 | |
| **Both MM2+ And PP5+** | 4306 | 14.3 | 80736 | 13.5 | 0.1 | |
| **Co-Occurrence of MM3+ and PP5+** | | | | | | |
| **Neither MM3+ Nor PP5+** | 15655 | 52.1 | 332521 | 55.6 | <0.01 | |
| **MM3+ Only** | 9950 | 33.1 | 129849 | 21.7 | <0.01 | |
| **PP5+ Only** | 581 | 1.9 | 84574 | 14.1 | <0.01 | |
| **Both MM3+ And PP5+** | 3911 | 13.0 | 50687 | 8.5 | <0.01 | |
| **Most Frequent Conditions and Medications** | | | | | | **Prevalence Difference (CLSA-CPCSSN)** |
| Hypertension | 11096 | **36.9** | 207649 | **34.7** | <0.01 | 2.1 |
| Obesity | 8933 | **29.7** | 256399 | **42.9** | <0.01 | -13.2 |
| Osteoarthritis or rheumatoid arthritis | 8434 | **28.0** | 82643 | **13.8** | <0.01 | 14.2 |
| Musculoskeletal problem | 8384 | **27.9** | 231404 | **38.7** | <0.01 | -10.9 |
| HMG CoA reductase inhibitors | 6320 | **21.0** | 74593 | **12.5** | <0.01 | 8.5 |
| Anxiety or depression | 6243 | **20.7** | 189064 | **31.6** | <0.01 | -10.9 |
| Diabetes | 5307 | **17.6** | 92954 | **15.6** | <0.01 | 2.1 |
| Chronic obstructive pulmonary disease or asthma | 5093 | **16.9** | 69575 | **11.6** | <0.01 | 5.3 |
| Cancer | 4636 | **15.4** | 146710 | **24.5** | <0.01 | -9.1 |
| Thyroid problem | 4376 | **14.5** | 58889 | **9.9** | <0.01 | 4.7 |
| Thyroid hormones | 3706 | **12.3** | 36050 | **6.0** | <0.01 | 6.3 |
| Proton pump inhibitors | 3523 | **11.7** | 80054 | **13.4** | <0.01 | -1.7 |
| Heart failure | 3503 | **11.6** | 10513 | 1.8 | <0.01 | 9.9 |
| Platelet aggregation inhibitors excl. heparin | 3359 | **11.2** | 33444 | **5.6** | <0.01 | 5.6 |
| ACE inhibitors, plain | 3030 | **10.1** | 48154 | **8.1** | <0.01 | 2.0 |
| Cardiovascular disease | 2867 | **9.5** | 58744 | **9.8** | <0.01 | -0.3 |
| Osteoporosis | 2688 | **8.9** | 8971 | 1.5 | <0.01 | 7.4 |
| Urinary problem | 2514 | **8.4** | 63918 | **10.7** | <0.01 | -2.3 |
| Stomach problem | 2275 | **7.6** | 31240 | **5.2** | <0.01 | 2.3 |
| Beta blocking agents, selective | 2166 | **7.2** | 33447 | **5.6** | <0.01 | 1.6 |
| Angiotensin II receptor blockers (ARBs) and diuretics | 2019 | **6.7** | 388 | 0.1 | <0.01 | 6.6 |

*(Continued)*

**Table 1.** (Continued)

| | CLSA (N = 30097) | | CPCSSN (N = 597631) | | p-value | |
|---|---|---|---|---|---|---|
| Dihydropyridine derivatives | 1784 | **5.9** | 31847 | **5.3** | 0.3 | 0.6 |
| Other antidepressants | 1760 | **5.8** | 48587 | **8.1** | <0.01 | -2.3 |
| Thiazides, plain | 1729 | **5.7** | 27759 | 4.6 | 0.04 | 1.1 |
| Selective serotonin reuptake inhibitors | 1570 | **5.2** | 55173 | **9.2** | <0.01 | -4.0 |
| Anilides | 1117 | 3.7 | 41022 | **6.9** | <0.01 | -3.2 |
| Propionic acid derivatives | 1050 | 3.5 | 56223 | **9.4** | <0.01 | -5.9 |
| Benzodiazepine derivatives | 986 | 3.3 | 59637 | **10.0** | <0.01 | -6.7 |
| Adrenergics in combination with corticosteroids or other drugs, excl. anticholinergics | 913 | 3.0 | 31903 | **5.3** | <0.01 | -2.3 |
| Biguanides | 901 | 3.0 | 31374 | **5.2** | <0.01 | -2.3 |
| Selective beta-2-adrenoreceptor agonists | 900 | 3.0 | 64945 | **10.9** | <0.01 | -7.9 |
| Glucocorticoids | 759 | 2.5 | 56990 | **9.5** | <0.01 | -7.0 |
| Corticosteroids | 718 | 2.4 | 87370 | **14.6** | <0.01 | -12.2 |
| Colon problem | 582 | 1.9 | 31081 | **5.2** | <0.01 | -3.3 |
| Natural opium alkaloids | 271 | 0.9 | 32766 | **5.5** | <0.01 | -4.6 |
| Corticosteroids, potent (group III) | 260 | 0.9 | 37299 | **6.2** | <0.01 | -5.4 |
| Progestogens and estrogens, fixed combinations | 84 | 0.3 | 40028 | **6.7** | 0.02 | -6.4 |
| Penicillins with extended spectrum | 42 | 0.1 | 38606 | **6.5** | 0.09 | -6.3 |
| Fluoroquinolones | 33 | 0.1 | 37953 | **6.4** | 0.14 | -6.2 |
| Macrolides | 26 | 0.1 | 53389 | **8.9** | 0.12 | -8.8 |

NB: CLSA participants were identified at baseline as of 2015 and CPCSSN patients were identified with any primary care encounter between 2014–2015; Only conditions and medications with an overall prevalence of at least 5.0% in either the CLSA or CPCSSN cohorts are presented, while the remaining conditions and medications are presented in S3 Table

## Middle-aged adults (45 to 64 years)

The results for the CLSA participants and CPCSSN patients aged 45 to 64 years are presented in Tables 2 and 3, respectively. Among CLSA participants, the overall prevalence of MM2 + was 57.7%, but female participants had a slightly higher prevalence of MM2+ as compared to males (61.7% and 53.5%, respectively). The prevalence of PP5+ was 10.0% and 9.1% among female and male CLSA participants aged 45 to 64 years, respectively. The highest proportion of CLSA participants were living with only MM2+ when categorized based on the co-occurrence of multimorbidity and polypharmacy (MM2+ and PP5+). The most frequent conditions among these CLSA participants were obesity, hypertension, musculoskeletal problem, anxiety or depression and osteoarthritis or rheumatoid arthritis. However, female CLSA participants had a higher prevalence of anxiety or depression (28.5%) as compared to male CLSA participants (18.1%) in this age group. There was more variation in the most frequent medications between females and males within this age group, such as a higher proportion of female participants taking thyroid hormones (14.5%) and higher proportion of male participants taking HMG CoA reductase inhibitors (17.7%).

Among CPCSSN patients aged 45 to 64 years, the prevalence of MM2+ and PP5+ was 44.0% and 18.1%, respectively. The proportion of CPCSSN patients in this age group who were categorized as having both MM2+ and PP5+ was 9.9%, which was comparable to 9.2% of CLSA participants in the same age group who were categorized as living with both MM2+ and PP5+. These similar proportions were despite the fact that the CLSA middle-aged participants

**Table 2. Prevalence and patterns of multimorbidity and polypharmacy in CLSA participants aged 45 to 64 years, overall and stratified by sex.**

| | All (n = 17451) | | Females (n = 9014) | | | Males (n = 8437) | | |
|---|---|---|---|---|---|---|---|---|
| **Mean Conditions (SD), Range** | 2.2 (1.8) | 0–15 | 2.4 (1.9) | 0–15 | | 2.0 (1.7) | 0–10 | |
| **Mean Medications (SD), Range** | 1.6 (2.1) | 0–20 | 1.7 (2.1) | 0–19 | | 1.5 (2.1) | 0–20 | |
| **Multimorbidity (MM)** | | | | | | | | |
| **Two or More Conditions (MM2+)** | 10075 | 57.7 | 5561 | 61.7 | | 4514 | 53.5 | |
| **Three or More Conditions (MM3+)** | 6303 | 36.1 | 3596 | 39.9 | | 2707 | 32.1 | |
| **Polypharmacy (PP)** | | | | | | | | |
| **Five or More Medications (PP5+)** | 1670 | 9.6 | 903 | 10.0 | | 767 | 9.1 | |
| **Ten or More Medications (PP10+)** | 156 | 0.9 | 84 | 0.9 | | 72 | 0.9 | |
| **Co-Occurrence of MM2+ and PP5+** | | | | | | | | |
| **Neither MM2+ Nor PP5+** | 7304 | 41.9 | 3417 | 37.9 | | 3887 | 46.1 | |
| **MM2+ Only** | 8477 | 48.6 | 4694 | 52.1 | | 3783 | 44.8 | |
| **PP5+ Only** | 72 | 0.4 | 36 | 0.4 | | 36 | 0.4 | |
| **Both MM2+ And PP5+** | 1598 | 9.2 | 867 | 9.6 | | 731 | 8.7 | |
| **Co-Occurrence of MM3+ and PP5+** | | | | | | | | |
| **Neither MM3+ Nor PP5+** | 10898 | 62.5 | 5293 | 58.7 | | 5605 | 66.4 | |
| **MM3+ Only** | 4883 | 28.0 | 2818 | 31.3 | | 2065 | 24.5 | |
| **PP5+ Only** | 250 | 1.4 | 125 | 1.4 | | 125 | 1.5 | |
| **Both MM3+ And PP5+** | 1420 | 8.1 | 778 | 8.6 | | 642 | 7.6 | |
| **Most Frequent Conditions** | | | | | | | | |
| Obesity | 5321 | 30.5 | Obesity | 2634 | 29.2 | Obesity | 2687 | 31.8 |
| Hypertension | 4856 | 27.8 | Anxiety or depression | 2569 | 28.5 | Hypertension | 2592 | 30.7 |
| Musculoskeletal problem | 4712 | 27.0 | Musculoskeletal problem | 2367 | 26.3 | Musculoskeletal problem | 2345 | 27.8 |
| Anxiety or depression | 4092 | 23.4 | Osteoarthritis or rheumatoid arthritis | 2276 | 25.2 | Anxiety or depression | 1523 | 18.1 |
| Osteoarthritis or rheumatoid arthritis | 3677 | 21.1 | Hypertension | 2264 | 25.1 | Osteoarthritis or rheumatoid arthritis | 1401 | 16.6 |
| **Most Frequent Medications** | | | | | | | | |
| HMG CoA reductase inhibitors | 2433 | 13.9 | Thyroid hormones | 1309 | 14.5 | HMG CoA reductase inhibitors | 1496 | 17.7 |
| Thyroid hormones | 1665 | 9.5 | HMG CoA reductase inhibitors | 937 | 10.4 | Platelet aggregation inhibitors excl. heparin | 804 | 9.5 |
| Proton pump inhibitors | 1569 | 9.0 | Proton pump inhibitors | 885 | 9.8 | ACE inhibitors, plain | 749 | 8.9 |
| Platelet aggregation inhibitors excl. heparin | 1183 | 6.8 | Other antidepressants | 771 | 8.6 | Proton pump inhibitors | 684 | 8.1 |
| ACE inhibitors, plain | 1172 | 6.7 | Natural and semisynthetic estrogens, plain | 740 | 8.2 | Beta blocking agents, selective | 397 | 4.7 |

had a higher prevalence of MM2+ (CLSA: 57.7%; CPCSSN: 44.0%) and the CPCSSN middle-aged patients had a higher prevalence of PP5+ (CLSA: 9.6%; CPCSSN: 18.1%). The most frequent conditions among CPCSSN patients were similar to the CLSA participants in the same age group, including obesity, hypertension, musculoskeletal problem and anxiety or depression. The most common medication among CPCSSN patients aged 45 to 64 years was corticosteroids (10.3%) for females and HMG CoA reductase inhibitors (8.5%) for males.

## Older adults (65 to 85 years)

The results for the CLSA participants and CPCSSN patients aged 65 to 85 years are presented in Tables 4 and 5, respectively. In this age group, the prevalence of MM2+ was 79.0% and the prevalence of PP5+ was 22.3% among CLSA participants, while the prevalence of MM2+ was

**Table 3. Prevalence and patterns of multimorbidity and polypharmacy in CPCSSN patients aged 45 to 64 years, overall and stratified by sex.**

| | All (n = 341732) | | Females (n = 189409) | | | Males (n = 152323) | | |
|---|---|---|---|---|---|---|---|---|
| Mean Conditions (SD), Range | 1.6 (1.4) | 0–14 | 1.6 (1.5) | 0–11 | | 1.5 (1.3) | 0–14 | |
| Mean Medications (SD), Range | 2.4 (4.2) | 0–80 | 2.6 (4.5) | 0–80 | | 2.1 (3.8) | 0–65 | |
| **Multimorbidity (MM)** | | | | | | | | |
| Two or More Conditions (MM2+) | 150483 | 44.0 | 87511 | 46.2 | | 62972 | 41.3 | |
| Three or More Conditions (MM3+) | 75306 | 22.0 | 45771 | 24.2 | | 29535 | 19.4 | |
| **Polypharmacy (PP)** | | | | | | | | |
| Five or More Medications (PP5+) | 61755 | 18.1 | 37743 | 19.9 | | 24012 | 15.7 | |
| Ten or More Medications (PP10+) | 22374 | 6.5 | 14596 | 7.7 | | 7778 | 5.1 | |
| **Co-Occurrence of MM2+ and PP5+** | | | | | | | | |
| Neither MM2+ Nor PP5+ | 163311 | 47.8 | 84879 | 44.8 | | 78432 | 51.5 | |
| MM2+ Only | 116655 | 34.1 | 66787 | 35.3 | | 49879 | 32.8 | |
| PP5+ Only | 27938 | 8.2 | 17019 | 9.0 | | 10919 | 7.2 | |
| Both MM2+ And PP5+ | 33817 | 9.9 | 20724 | 10.9 | | 13093 | 8.6 | |
| **Co-Occurrence of MM3+ and PP5+** | | | | | | | | |
| Neither MM3+ Nor PP5+ | 223673 | 65.5 | 117829 | 62.2 | | 105844 | 69.5 | |
| MM3+ Only | 56304 | 16.5 | 33837 | 17.9 | | 22467 | 14.8 | |
| PP5+ Only | 42753 | 12.5 | 25809 | 13.6 | | 16944 | 11.1 | |
| Both MM3+ And PP5+ | 19002 | 5.5 | 11934 | 6.3 | | 7068 | 4.6 | |
| **Most Frequent Conditions** | | | | | | | | |
| Obesity | 101250 | 29.6 | Obesity | 54922 | 29.0 | Obesity | 46328 | 30.4 |
| Musculoskeletal problem | 92101 | 27.0 | Musculoskeletal problem | 52537 | 27.7 | Musculoskeletal problem | 39564 | 26.0 |
| Anxiety or depression | 68848 | 20.1 | Anxiety or depression | 44764 | 23.6 | Hypertension | 33999 | 22.3 |
| Hypertension | 66117 | 19.3 | Cancer | 33514 | 17.7 | Anxiety or depression | 24084 | 15.8 |
| Cancer | 50316 | 14.7 | Hypertension | 32118 | 17.0 | Cancer | 16802 | 11.0 |
| **Most Frequent Medications** | | | | | | | | |
| Corticosteroids | 31851 | 9.3 | Corticosteroids | 19563 | 10.3 | HMG CoA reductase inhibitors | 13016 | 8.5 |
| Proton pump inhibitors | 27460 | 8.0 | Proton pump inhibitors | 15693 | 8.3 | Corticosteroids | 12288 | 8.1 |
| Propionic acid derivatives | 24123 | 7.1 | Propionic acid derivatives | 14496 | 7.7 | Proton pump inhibitors | 11767 | 7.7 |
| Benzodiazepine derivatives | 21709 | 6.4 | Benzodiazepine derivatives | 14443 | 7.6 | Propionic acid derivatives | 9627 | 6.3 |
| Selective beta-2-adrenoreceptor agonists | 21433 | 6.3 | Selective beta-2-adrenoreceptor agonists | 13430 | 7.1 | ACE inhibitors, plain | 8136 | 5.3 |

62.6% and the prevalence of PP5+ was 28.7% among CPCSSN patients. Approximately 20.0% of CLSA participants and CPCSSN patients in this age group were identified as living with both MM2+ and PP5+ (CLSA: 21.4%; CPCSSN: 18.3%). The most common conditions among the CLSA participants (overall and stratified by sex) were hypertension (49.3%) and osteoarthritis or rheumatoid arthritis (37.6%). In comparison, the most common conditions among CPCSSN patients (overall and stratified by sex) were hypertension (43.5%) and obesity (30.4%). The most common medication for the older adults in the CLSA and CPCSSN cohorts, including when stratified between females and males, was HMG CoA reductase inhibitors (CLSA: 30.7%; CPCSSN: 18.1%).

## Discussion

### Summary of results

When defined as two or more conditions, the overall prevalence of multimorbidity was 66.7% in the CLSA cohort and 52.0% in the CPCSSN cohort. When defined as five or more medications, the overall prevalence of polypharmacy was 14.9% in the CLSA cohort and 22.6% in the

**Table 4. Prevalence and patterns of multimorbidity and polypharmacy in CLSA participants aged 65 to 85 years, overall and stratified by sex.**

| | All (n = 12646) | | Females (n = 6306) | | | Males (n = 6340) | | |
|---|---|---|---|---|---|---|---|---|
| **Mean Conditions (SD), Range** | 3.3 (2.1) | 0–13 | 3.5 (2.1) | | 0–13 | 3.0 (2.0) | | 0–12 |
| **Mean Medications (SD), Range** | 2.9 (2.5) | 0–21 | 2.9 (2.5) | | 0–21 | 2.9 (2.5) | | 0–18 |
| **Multimorbidity (MM)** | | | | | | | | |
| **Two or More Conditions (MM2+)** | 9985 | 79.0 | 5220 | | 82.8 | 4765 | | 75.2 |
| **Three or More Conditions (MM3+)** | 7558 | 59.8 | 4100 | | 65.0 | 3458 | | 54.5 |
| **Polypharmacy (PP)** | | | | | | | | |
| **Five or More Medications (PP5+)** | 2822 | 22.3 | 1391 | | 22.1 | 1431 | | 22.6 |
| **Ten or More Medications (PP10+)** | 249 | 2.0 | 127 | | 2.0 | 122 | | 1.9 |
| **Co-Occurrence of MM2+ and PP5+** | | | | | | | | |
| **Neither MM2+ Nor PP5+** | 2547 | 20.1 | 1050 | | 16.7 | 1497 | | 23.6 |
| **MM2+ Only** | 7277 | 57.5 | 3865 | | 61.3 | 3412 | | 53.8 |
| **PP5+ Only** | 114 | 0.9 | 36 | | 0.6 | 78 | | 1.2 |
| **Both MM2+ And PP5+** | 2708 | 21.4 | 1355 | | 21.5 | 1353 | | 21.3 |
| **Co-Occurrence of MM3+ and PP5+** | | | | | | | | |
| **Neither MM3+ Nor PP5+** | 4757 | 37.6 | 2089 | | 33.1 | 2668 | | 42.1 |
| **MM3+ Only** | 5067 | 40.1 | 2826 | | 44.8 | 2241 | | 35.4 |
| **PP5+ Only** | 331 | 2.6 | 117 | | 1.9 | 214 | | 3.4 |
| **Both MM3+ And PP5+** | 2491 | 19.7 | 1274 | | 20.2 | 1217 | | 19.2 |
| **Most Frequent Conditions** | | | | | | | | |
| Hypertension | 6240 | 49.3 | Hypertension | 3106 | 49.3 | Hypertension | 3134 | 49.4 |
| Osteoarthritis or rheumatoid arthritis | 4757 | 37.6 | Osteoarthritis or rheumatoid arthritis | 2934 | 46.5 | Osteoarthritis or rheumatoid arthritis | 1823 | 28.8 |
| Musculoskeletal problem | 3672 | 29.0 | Obesity | 1890 | 30.0 | Musculoskeletal problem | 1810 | 28.5 |
| Obesity | 3612 | 28.6 | Musculoskeletal problem | 1862 | 29.5 | Obesity | 1722 | 27.2 |
| Cancer | 2939 | 23.2 | Thyroid problem | 1663 | 26.4 | Cancer | 1584 | 25.0 |
| **Most Frequent Medications** | | | | | | | | |
| HMG CoA reductase inhibitors | 3887 | 30.7 | HMG CoA reductase inhibitors | 1520 | 24.1 | HMG CoA reductase inhibitors | 2367 | 37.3 |
| Platelet aggregation inhibitors excl. heparin | 2176 | 17.2 | Thyroid hormones | 1458 | 23.1 | Platelet aggregation inhibitors excl. heparin | 1352 | 21.3 |
| Thyroid hormones | 2041 | 16.1 | Proton pump inhibitors | 1078 | 17.1 | ACE inhibitors, plain | 1149 | 18.1 |
| Proton pump inhibitors | 1954 | 15.5 | Platelet aggregation inhibitors excl. heparin | 824 | 13.1 | Beta blocking agents, selective | 940 | 14.8 |
| ACE inhibitors, plain | 1858 | 14.7 | ACE inhibitors, plain | 709 | 11.2 | Proton pump inhibitors | 876 | 13.8 |

CPCSSN cohort. Using these same two definitions, the co-occurrence of MM2+ and PP5+ was 14.3% among CLSA participants and 13.5% among CPCSSN patients. When stratified between age groups (45 to 64 years and 65 to 85 years), the co-occurrence of MM2+ and PP5+ was lower in middle-aged adults (CLSA: 9.2%; CPCSSN: 9.9%) as compared to older adults (CLSA: 21.4%; CPCSSN: 18.3%). As seen in Fig 1, the proportion of those with this co-occurrence is much smaller than those living with only multimorbidity. While polypharmacy without multimorbidity is rare, multimorbidity without polypharmacy is common. The prevalence of multimorbidity and polypharmacy was consistently higher among females as compared to males in both datasets and across both age groups. However, these differences between females and males were minor as the largest difference in prevalence estimates was observed among CLSA participants aged 65 to 85 years when reporting the prevalence of three or more conditions (females: 65.0%; males: 54.5%).

**Table 5. Prevalence and patterns of multimorbidity and polypharmacy in CPCSSN patients aged 65 to 85 years, overall and stratified by sex.**

| | All (n = 255899) | | Females (n = 139332) | | | Males (n = 116567) | | |
|---|---|---|---|---|---|---|---|---|
| Mean Conditions (SD), Range | 2.3 (1.8) | 0–14 | 2.4 (1.8) | 0–14 | | 2.3 (1.7) | 0–13 | |
| Mean Medications (SD), Range | 3.8 (6.0) | 0–88 | 3.9 (6.2) | 0–88 | | 3.6 (5.7) | 0–68 | |
| **Multimorbidity (MM)** | | | | | | | | |
| Two or More Conditions (MM2+) | 160261 | 62.6 | 88115 | 63.2 | | 72146 | 61.9 | |
| Three or More Conditions (MM3+) | 105230 | 41.1 | 58820 | 42.2 | | 46410 | 39.8 | |
| **Polypharmacy (PP)** | | | | | | | | |
| Five or More Medications (PP5+) | 73506 | 28.7 | 40871 | 29.3 | | 32635 | 28.1 | |
| Ten or More Medications (PP10+) | 36447 | 14.2 | 20933 | 15.0 | | 15514 | 13.4 | |
| **Co-Occurrence of MM2+ and PP5+** | | | | | | | | |
| Neither MM2+ Nor PP5+ | 69051 | 27.0 | 36521 | 26.2 | | 32530 | 27.9 | |
| MM2+ Only | 113342 | 44.3 | 61940 | 44.5 | | 51402 | 44.1 | |
| PP5+ Only | 26587 | 10.4 | 14696 | 10.5 | | 11891 | 10.2 | |
| Both MM2+ And PP5+ | 46919 | 18.3 | 26175 | 18.8 | | 20744 | 17.8 | |
| **Co-Occurrence of MM3+ and PP5+** | | | | | | | | |
| Neither MM3+ Nor PP5+ | 108848 | 42.5 | 57547 | 41.3 | | 51301 | 44.0 | |
| MM3+ Only | 73545 | 28.7 | 40914 | 29.4 | | 32631 | 28.0 | |
| PP5+ Only | 41821 | 16.3 | 22965 | 16.5 | | 18856 | 16.2 | |
| Both MM3+ And PP5+ | 31685 | 12.4 | 17906 | 12.8 | | 13779 | 11.8 | |
| **Most Frequent Conditions** | | | | | | | | |
| Hypertension | 111233 | 43.5 | Hypertension | 60037 | 43.1 | Hypertension | 51196 | 43.9 |
| Obesity | 77802 | 30.4 | Obesity | 41374 | 29.7 | Obesity | 36428 | 31.3 |
| Musculoskeletal problem | 70426 | 27.5 | Musculoskeletal problem | 40531 | 29.1 | Musculoskeletal problem | 29895 | 25.6 |
| Cancer | 53935 | 21.1 | Cancer | 29334 | 21.1 | Diabetes | 26990 | 23.2 |
| Diabetes | 50452 | 19.7 | Osteoarthritis or rheumatoid arthritis | 28602 | 20.5 | Cancer | 24601 | 21.1 |
| **Most Frequent Medications** | | | | | | | | |
| HMG CoA reductase inhibitors | 46350 | 18.1 | HMG CoA reductase inhibitors | 21312 | 15.3 | HMG CoA reductase inhibitors | 25038 | 21.5 |
| Proton pump inhibitors | 32242 | 12.6 | Proton pump inhibitors | 18682 | 13.4 | ACE inhibitors, plain | 15290 | 13.1 |
| ACE inhibitors, plain | 27494 | 10.7 | Corticosteroids | 14192 | 10.2 | Proton pump inhibitors | 13560 | 11.6 |
| Corticosteroids | 23861 | 9.3 | Benzodiazepine derivatives | 13671 | 9.8 | Platelet aggregation inhibitors excl. heparin | 12239 | 10.5 |
| Platelet aggregation inhibitors excl. heparin | 21169 | 8.3 | Glucocorticoids | 12394 | 8.9 | Beta blocking agents, selective | 11154 | 9.6 |

The comparative distributions of the CLSA and CPCSSN cohorts between the four mutually exclusive categories based on the co-occurrence of multimorbidity and polypharmacy (neither MM nor PP, MM only, PP only and both MM and PP), stratified by age group (45 to 64 years and 65 to 85 years) and by sex (female and male), are presented in Fig 1 (for MM2+ and PP5+) and Fig 2 (for MM3+ and PP5+).

The differences in the prevalence of multimorbidity and polypharmacy between the CLSA and CPCSSN cohorts may be due to many factors including differences in the sample (community-dwelling vs primary care) or differences in the way that data were captured (self-reported vs diagnostic codes). Despite the differences in the data collection methods and the moderate differences in the overall prevalence of multimorbidity and polypharmacy between the CLSA and CPCSSN cohorts individually, there were still consistencies in the co-occurrence of multimorbidity and polypharmacy, which was found to be approximately one in five older adults in both cohorts. Individuals living with multimorbidity do not necessarily have

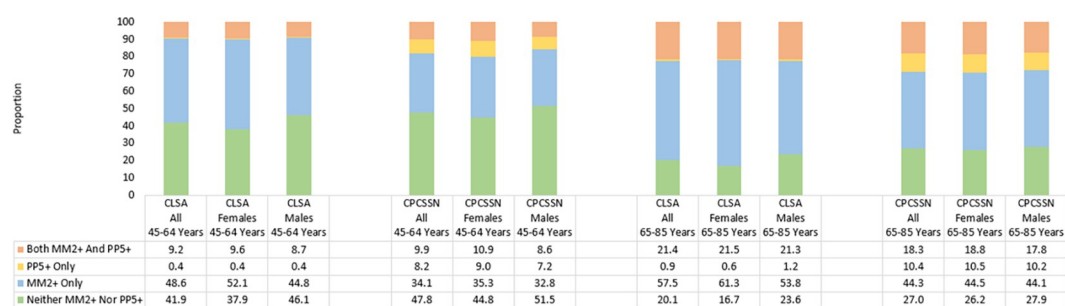

**Fig 1. Prevalence of multimorbidity (two or more conditions) and polypharmacy (five or more medications) in CLSA and CPCSSN cohorts, stratified by age group and sex.**

polypharmacy, while those living with polypharmacy almost invariably have multimorbidity. Our results show that multimorbidity and polypharmacy are not interchangeable in understanding population health needs. Indeed, those living with both multimorbidity and polypharmacy likely represents an important group for focus in both the approach to clinical practice and health policy.

## Findings in context

Differences still exist between studies in the lists of individual conditions and medications that are used in the operational definitions of these concepts and this has an impact on resulting prevalence statistics [57]. As such, comparisons were made with studies that utilized the same cut-points for multimorbidity (two or more) and polypharmacy (five or more) as used in the current study. The prevalence of multimorbidity in the Canadian Chronic Disease Surveillance System (CCDSS) in 2015–2016 was found to be 32.4% among community-dwelling adults aged 35 years and older [80]. In the Canadian Community Health Survey (CCHS), the prevalence of multimorbidity in 2017–2018 was approximately 34.1% of community-dwelling adults aged 65 years and older [81]. Both of these estimates of multimorbidity were lower than the prevalence of multimorbidity in the current study. For the prevalence of polypharmacy, the Canadian Health Measures Survey (CHMS) in 2016–2017 found that 12.6% of community-dwelling adults aged 40 to 59 years and 32.1% of community-dwelling adults 60 to 79 years were taking five or more prescription medications [82], which were comparable to the prevalence of polypharmacy in the current study. The Canadian Institutes for Health Information

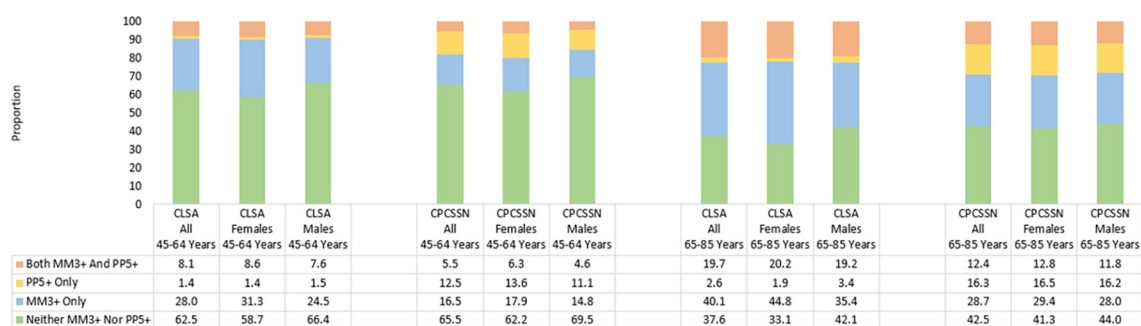

**Fig 2. Prevalence of multimorbidity (three or more conditions) and polypharmacy (five or more medications) in CLSA and CPCSSN cohorts, stratified by age group and sex.**

(CIHI) in 2016 found that 65.7% of adults aged 65 years and older in Canada were prescribed five or more medication classes, which was substantially higher than the percentages reported in the current study as this administrative dataset included those living in long-term care settings [83].

In the context of other national health survey datasets, the CLSA cohort had a higher prevalence of multimorbidity and a lower prevalence of polypharmacy as compared to studies of the Longitudinal Aging Study Amsterdam (LASA) where multimorbidity prevalence was 45.3% and polypharmacy prevalence was 36.8% among participants aged 65 years and older [84]; the Irish Longitudinal Study on Ageing (TILDA) where multimorbidity prevalence was 59.3% and polypharmacy prevalence was 20.8% among participants aged 50 years and older [85]; and the Survey of Health, Ageing and Retirement in Europe (SHARE) where multimorbidity prevalence was 49.6% and polypharmacy prevalence was 22.3% among participants aged 50 years and older living in the eighteen countries that participated in wave 6 data collection [86]. In contrast, the prevalence of polypharmacy in the CLSA cohort was similar to that reported using the Brazilian Longitudinal Study of Adult Health (ELSA-Brasil), which was 11.7% among participants who were 35 to 74 years of age [87]. However, it is challenging to articulate if the differences in the prevalence estimates of multimorbidity and polypharmacy that are observed between these cohorts represent true differences or represent the impact of varying lists of conditions and medications.

In comparison with other large primary care datasets, the prevalence of both multimorbidity and polypharmacy in the CPCSSN cohort was comparable, but slightly lower than the prevalence reported by Sinnige et al. [88] using the NIVEL Primary Care Database (NIVEL-PCD) in the Netherlands. Amongst 45731 patients aged 55 years and older who received primary care from participating practices, the prevalence of multimorbidity was 58.0% and the prevalence of polypharmacy was 27.0% [88]. Another study in the Netherlands by van den Akker et al. [89] used Intego, a Flemish-Belgian general practice-based morbidity registration network at the Academic Centre of General Practice of the KU Leuven, which found a lower prevalence of multimorbidity (46.5%) and a higher prevalence of polypharmacy (40.7%) than the current study amongst primary care patients aged 50 years and older. In contrast, a study by O'Regan et al. [90] using EMR data in Ireland among patients aged 50 years and older who received primary care reported a lower prevalence of multimorbidity (37.6%), but a higher prevalence of polypharmacy (38.5%) than those estimates reported from the CPCSSN cohort. This study examined prevalence among 6603 patients from 68 general practices associated with the University of Limerick Education and Research Network for General Practice (ULEARN-GP) [90]. Interestingly, a higher prevalence of polypharmacy than multimorbidity is not common in other literature and was not seen in the results of the current study.

## Strengths and limitations

The strengths of this study include the application of a consistent definition and cut-point in identifying multimorbidity and polypharmacy across two national datasets. Although this required modification of a previously used definition of multimorbidity by Fortin et al. [59] as not all 20 conditions in the original multimorbidity definition were captured in the CLSA data, this consistent methodology supported the comparison between the two cohorts (including same age groups between samples). It was also important to present age- and sex-stratified analyses (gender was not captured in the CPCSSN data) to articulate differences between these groups as both multimorbidity and polypharmacy are known to be influenced by age, sex and gender [91, 92]. This study presents one of the first reports of the co-occurrence of multimorbidity and polypharmacy (in addition to reporting on the individual prevalence estimates)

using two large and representative datasets in Canada, which can continue to be used for longitudinal surveillance of these two significant health states. In addition to facilitating international comparisons, large population-based datasets should play a key role in reporting on the changing occurrence of multimorbidity and polypharmacy over time, as well as contributing to the practice of pharmacovigilance [93].

The limitations of this study include the fact that multimorbidity and polypharmacy estimates were only stratified using age and sex categories due to the lack of more comprehensive information in the CPCSSN EMR database. The CLSA dataset contains a broad spectrum of extensive variables related to health and well-being (such as socioeconomic factors and health behaviours), but similar variables were not available in the CPCSSN dataset [73, 77]. Ideally, more comprehensive data should be entered and extracted from EMRs for clinical and research purposes. The medications data in the CPCSSN dataset were also included if there were no stop dates documented with the intent to capture "active" medications. However, this field may not be consistently documented by the primary care provider and the prescribed medications may have been stopped even if a stop date was not entered into the EMR. As well, the CLSA contains self-reported use of natural and non-prescribed products, which were not captured in the CPCSSN dataset, but can contribute to the complexity of a treatment regimen and should be explored in future research. Finally, although the data collection sites for both the CLSA and CPCSSN are located in many of the same cities in Canada, there is no current linkage to identify participants and patients who may be in both datasets, but there would be significant benefit of linking these two datasets for future research (including the validation between self-reported and clinically documented health conditions and current medications).

## Policy and practice implications

The presence of multimorbidity and polypharmacy can have substantial impact on the health status, health outcomes and health care use by individuals. This study emphasizes the need for a shift away from single disease clinical guidelines, toward clinical practice pathways that account for the co-occurrence of multiple conditions and treatments. Data from this study indicate that this is particularly important for individuals aged 65 years and older, as the proportion with co-occurrence was double that amongst middle-aged adults. The magnitude of benefits and side effects also shift with increasing age, as can individual preferences and priorities for treatment. A starting point for the shift away from single disease approaches would be to develop a completely different approach and separate pathways for this age group. This would support and guide discussions around the appropriate use of medications and other non-pharmacological interventions to facilitate effective management, as well as to reduce treatment burden [3, 4, 17]. While population level data cannot assess individual appropriateness of polypharmacy, multimorbidity does not have to result in substantial or problematic polypharmacy if clinicians are judicious in rational prescribing. This can be achieved through approaches to unwanted or unnecessary medications such as deprescribing, eliminating legacy prescribing (when medications that were appropriate at initiation for an intermediate term are not appropriately discontinued and subsequently lead to adverse outcomes) [18], using medications that can treat multiple co-existing conditions, reducing doses or discontinuing medications or combinations where the risks are too great [3, 94]. Health policy needs to take this broader view when considering resourcing, defining quality of care and related measurement and incentives. Practice and policy should continue to emphasize the value of patient-centered and team-focused approaches to providing comprehensive health care.

## Future research

This study did not examine any patient-important outcomes of concurrent multimorbidity and polypharmacy (such as quality of life, treatment burden or adverse events), but future research should continue to examine these outcomes and their occurrence in the context of independent and concurrent multimorbidity and polypharmacy. More nuanced information can come from assessing the impact, both separately and combined, as well as examining sub-analyses of patterns and combinations of morbidities, risk factors and related medications. Even further, research should continue to examine the impact of socioeconomic status and lifestyle behaviours, although these were not explored in the current study. While pharmaceutical management of risk factors can add to treatment burden, this is also a useful proxy marker for the severity of the condition, such as hypertension. Further analysis of related health service utilization and health system costs can provide useful information for policy and planning. Ideally, future research should leverage multiple sources of data to triangulate information from different perspectives, as we have in this study, which can provide insights into the potential for prevention and intervention at clinical and population levels.

## Conclusion

This study reported on the prevalence of the co-occurrence of multimorbidity and polypharmacy among middle-aged and older adults in two national datasets in Canada. Each dataset has strengths and limitations that can be leveraged in interpretation, and so combined the CLSA and CPCSSN datasets provide a more comprehensive assessment of these health states from both a population and primary care perspective. To inform health policy and clinical practice, future research and routine analyses should report on the co-occurrence of multimorbidity and polypharmacy. This insight can be used to further examine the impact and opportunities of interventions, evidence and clinical pathways that are tailored to individuals who are living with multimorbidity and polypharmacy in order to improve health outcomes over time.

## Supporting information

**S1 Table. List of conditions (ICD-9) for definition of multimorbidity.**
(PDF)

**S2 Table. List of medications (ATC Level 4) for definition of polypharmacy.**
(PDF)

**S3 Table. Prevalence of conditions and medications in CLSA and CPCSSN.**
(PDF)

## Acknowledgments

This study was made possible using the data/biospecimens collected by the Canadian Longitudinal Study on Aging (CLSA) and the Canadian Primary Care Sentinel Surveillance Network (CPCSSN). This study was conducted using the CLSA Baseline Comprehensive Dataset Version 6.0 under Application Number 2104064 and the CPCSSN Dataset under Application Number 2017SRSC99. The CLSA is led by Drs. Parminder Raina, Christina Wolfson and Susan Kirkland.

## Author Contributions

**Conceptualization:** Kathryn Nicholson, Dee Mangin.

**Data curation:** Kathryn Nicholson.

**Formal analysis:** Kathryn Nicholson.

**Funding acquisition:** Kathryn Nicholson, Dee Mangin.

**Methodology:** Kathryn Nicholson.

**Writing – original draft:** Kathryn Nicholson.

**Writing – review & editing:** Kathryn Nicholson, Jennifer Salerno, Sayem Borhan, Benoit Cossette, Dale Guenter, Meredith Vanstone, John Queenan, Michelle Greiver, Michelle Howard, Amanda L. Terry, Tyler Williamson, Lauren E. Griffith, Martin Fortin, Saverio Stranges, Dee Mangin.

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
