## [Decision Letter · Decision Letter 0]

7 Aug 2024

PONE-D-24-26485The co-occurrence of multimorbidity and polypharmacy among middle-aged and older adults in Canada: A cross-sectional study using the Canadian Longitudinal Study on Aging (CLSA) and the Canadian Primary Care Sentinel Surveillance Network (CPCSSN)PLOS ONE

Dear Dr. Nicholson,

Thank you for submitting your manuscript to PLOS ONE. After careful consideration, we feel that it has merit but does not fully meet PLOS ONE’s publication criteria as it currently stands. Therefore, we invite you to submit a revised version of the manuscript that addresses the points raised during the review process.

We look forward to receiving your revised manuscript.

Kind regards,

Sima Afrashteh

Academic Editor

PLOS ONE

Journal Requirements:

This study was supported by the Canadian Institutes of Health Research Priority Announcement in Aging (175365).  This funding source had no role in the conduct of this study, the writing of the manuscript or the decision to submit for publication.  

This study was made possible using the data/biospecimens collected by the Canadian Longitudinal Study on Aging (CLSA) and the Canadian Primary Care Sentinel Surveillance Network (CPCSSN).  Funding for the CLSA is provided by the Government of Canada through the Canadian Institutes of Health Research (CIHR) under grant reference: LSA 94473 and the Canada Foundation for Innovation, as well as the following provinces, Newfoundland, Nova Scotia, Quebec, Ontario, Manitoba, Alberta, and British Columbia.  This study was conducted using the CLSA Baseline Comprehensive Dataset Version 6.0 under Application Number 2104064 and the CPCSSN Dataset under Application Number 2017SRSC99.  The CLSA is led by Drs. Parminder Raina, Christina Wolfson and Susan Kirkland.  The research ethics approval was obtained from the Hamilton Integrated Research Ethics Board (HiREB) Project Number 3586.  This study was supported by the Canadian Institutes of Health Research Priority Announcement in Aging (175365).  This funding source had no role in the conduct of this study, the writing of the manuscript or the decision to submit for publication.  There were no other funding sources for this research. 

This study was supported by the Canadian Institutes of Health Research Priority Announcement in Aging (175365).  This funding source had no role in the conduct of this study, the writing of the manuscript or the decision to submit for publication.  

4. Thank you for uploading your study's underlying data set. Unfortunately, the repository you have noted in your Data Availability statement does not qualify as an acceptable data repository according to PLOS's standards.

7. Please remove your figures from within your manuscript file, leaving only the individual TIFF/EPS image files, uploaded separately. These will be automatically included in the reviewers’ PDF.

Reviewers' comments:

Reviewer's Responses to Questions

**Comments to the Author**

1. Is the manuscript technically sound, and do the data support the conclusions?

Reviewer #1: Yes

Reviewer #2: Yes

2. Has the statistical analysis been performed appropriately and rigorously? 

Reviewer #1: No

Reviewer #2: Yes

3. Have the authors made all data underlying the findings in their manuscript fully available?

Reviewer #1: No

Reviewer #2: Yes

4. Is the manuscript presented in an intelligible fashion and written in standard English?

Reviewer #1: Yes

Reviewer #2: Yes

5. Review Comments to the Author

**Reviewer #1: **The submitted manuscript presents valuable evidence on the prevalence of polypharmacy (PP) and multimorbidity (MM) on a countrywide scale in Canada. It is authored properly in standard English and is properly structured as well. However, two points should be taken into consideration. First, concerning the abstract, the maximum permitted length is 300 words (provided with ~ 380). Second, as stated in the data resource profile of the CPCSSN (doi: 10.1093/ije/dyw248) “… older adults and females are over-represented in the CPCSSN database as compared with the national population. As such it is important to consider age and sex standardization and/or adjustment for all surveillance and research studies employing these data”, it is advisable also to report the age- and sex-standardized values of the overall prevalence of the co-occurrence of MM and PP (and each in isolation) for this dataset, or to state the reason(s) for the exclusion of this step.

Restrictions on publicly sharing data: third-party data

**Reviewer #2:** Dear Author, Thank you for conducting this study. However, The following points should be considered:

1- The abstract is relatively long and could be more concise, particularly in the results section, to enhance readability and clarity.

2- Although the authors stated that they conducted a descriptive study, a statistical comparison (e.g., Chi-squared test in Table 1) could be useful for better understanding the differences between the two cohorts.

3- The discussion section could be strengthened by addressing the following points:

A. The reasons for the differences in the prevalence of poly-pharmacy and multi-morbidity between studies.

B. The clinical consequences and health impacts of poly-pharmacy and multi-morbidity should be addressed.

C. Discussing the influence of factors such as socioeconomic status and lifestyle on poly-pharmacy and multi-morbidity can enhance understanding of the topic and the significance of the study.

D. The discussion should align with the results and emphasize the significance and effectiveness of such studies.

6. PLOS authors have the option to publish the peer review history of their article (what does this mean?). If published, this will include your full peer review and any attached files.

Reviewer #1: **Yes: **Shahrokh Mousavi

Reviewer #2: No

---

## [Author Response · Author response to Decision Letter 0]

20 Sep 2024

RESPONSE: The style requirements were reviewed and updated as requested. 

This study was supported by the Canadian Institutes of Health Research Priority Announcement in Aging (175365). This funding source had no role in the conduct of this study, the writing of the manuscript or the decision to submit for publication. 

RESPONSE: The funding statement was updated as requested. 

This study was made possible using the data/biospecimens collected by the Canadian Longitudinal Study on Aging (CLSA) and the Canadian Primary Care Sentinel Surveillance Network (CPCSSN). Funding for the CLSA is provided by the Government of Canada through the Canadian Institutes of Health Research (CIHR) under grant reference: LSA 94473 and the Canada Foundation for Innovation, as well as the following provinces, Newfoundland, Nova Scotia, Quebec, Ontario, Manitoba, Alberta, and British Columbia. This study was conducted using the CLSA Baseline Comprehensive Dataset Version 6.0 under Application Number 2104064 and the CPCSSN Dataset under Application Number 2017SRSC99. The CLSA is led by Drs. Parminder Raina, Christina Wolfson and Susan Kirkland. The research ethics approval was obtained from the Hamilton Integrated Research Ethics Board (HiREB) Project Number 3586. This study was supported by the Canadian Institutes of Health Research Priority Announcement in Aging (175365). This funding source had no role in the conduct of this study, the writing of the manuscript or the decision to submit for publication. There were no other funding sources for this research. 

This study was supported by the Canadian Institutes of Health Research Priority Announcement in Aging (175365). This funding source had no role in the conduct of this study, the writing of the manuscript or the decision to submit for publication. 

RESPONSE: The acknowledgements section was updated as requested. 

4. Thank you for uploading your study's underlying data set. Unfortunately, the repository you have noted in your Data Availability statement does not qualify as an acceptable data repository according to PLOS's standards.

RESPONSE: Unfortunately, the data cannot be reposited as requested based on the guidelines of the CLSA and CPCSSN datasets, but the Data Availability Statement outlines the process for obtaining access to the respective datasets. 

RESPONSE: The ORCID iD for the corresponding authors was added as requested. 

RESPONSE: The ethics statement was included in the Methods section as requested. 

7. Please remove your figures from within your manuscript file, leaving only the individual TIFF/EPS image files, uploaded separately. These will be automatically included in the reviewers’ PDF. 

RESPONSE: The figures were uploaded separately as requested. 

RESPONSE: The reference list was reviewed as requested. 

Reviewer #1: 

1. First, concerning the abstract, the maximum permitted length is 300 words (provided with ~ 380).

RESPONSE: The abstract has been updated as requested. 

2. Second, as stated in the data resource profile of the CPCSSN (doi: 10.1093/ije/dyw248) “… older adults and females are over-represented in the CPCSSN database as compared with the national population. As such it is important to consider age and sex standardization and/or adjustment for all surveillance and research studies employing these data”, it is advisable also to report the age- and sex-standardized values of the overall prevalence of the co-occurrence of MM and PP (and each in isolation) for this dataset, or to state the reason(s) for the exclusion of this step. 

RESPONSE: The prevalence of the co-occurrence of multimorbidity and polypharmacy has been presented stratified between age groups (middle aged and older adults) and between females and males. 

Reviewer #2: 

1. The abstract is relatively long and could be more concise, particularly in the results section, to enhance readability and clarity. 

RESPONSE: The abstract has been updated as requested. 

2. Although the authors stated that they conducted a descriptive study, a statistical comparison (e.g., Chi-squared test in Table 1) could be useful for better understanding the differences between the two cohorts. 

RESPONSE: The statistical comparison in Table 1 was provided as requested. 

3. The discussion section could be strengthened by addressing the following points:

A. The reasons for the differences in the prevalence of poly-pharmacy and multi-morbidity between studies.

B. The clinical consequences and health impacts of poly-pharmacy and multi-morbidity should be addressed.

C. Discussing the influence of factors such as socioeconomic status and lifestyle on poly-pharmacy and multi-morbidity can enhance understanding of the topic and the significance of the study.

D. The discussion should align with the results and emphasize the significance and effectiveness of such studies. 

RESPONSE: The discussion section was updated with these details as requested.

---

## [Decision Letter · Decision Letter 1]

15 Oct 2024

The co-occurrence of multimorbidity and polypharmacy among middle-aged and older adults in Canada: a cross-sectional study using the Canadian Longitudinal Study on Aging (CLSA) and the Canadian Primary Care Sentinel Surveillance Network (CPCSSN)

PONE-D-24-26485R1

Dear Dr. Nicholson,

We’re pleased to inform you that your manuscript has been judged scientifically suitable for publication and will be formally accepted for publication once it meets all outstanding technical requirements.

Kind regards,

Sima Afrashteh

Academic Editor

PLOS ONE

---

## [Editor Report · Acceptance letter]

12 Nov 2024

PONE-D-24-26485R1 

PLOS ONE

Dear Dr. Nicholson, 

I'm pleased to inform you that your manuscript has been deemed suitable for publication in PLOS ONE. Congratulations! Your manuscript is now being handed over to our production team.

Kind regards, 

on behalf of

Dr. Sima Afrashteh 

Academic Editor

PLOS ONE